# Integrated analysis of DNA methylation profile of *HLA-G* gene and imaging in coronary heart disease: Pilot study

**Concetta Schiano**[1☯]*, **Giuditta Benincasa**[1☯], **Teresa Infante**[1], **Monica Franzese**[2], **Rossana Castaldo**[2], **Carmela Fiorito**[3], **Gelsomina Mansueto**[1], **Vincenzo Grimaldi**[4], **Giovanni Della Valle**[5], **Gerardo Fatone**[5], **Andrea Soricelli**[2,6], **Giovanni Francesco Nicoletti**[7], **Antonio Ruocco**[8], **Ciro Mauro**[8], **Marco Salvatore**[2], **Claudio Napoli**[1,2]

1 Department of Advanced Medical and Surgical Sciences (DAMSS), University of Campania "L. Vanvitelli", Naples, Italy, 2 IRCCSSDN, Naples, Italy, 3 U.O.C. Division of Clinical Immunology, Immunohematology, Transfusion Medicine and Transplant Immunology [SIMT], University of Campania "L. Vanvitelli", Naples, Italy, 4 Division of Clinical Immunology, Immunohematology, Transfusion Medicine and Transplant Immunology [SIMT], Clinical Department of Internal Medicine and Specialistic Units, Regional Reference Laboratory of Transplant Immunology [LIT], Azienda Universitaria Policlinico (AOU), Naples, Italy, 5 Department of Veterinary Medicine and Animal Production, University of Napoli Federico II, Napoli, Italy, 6 Department of Exercise and Wellness Sciences, University of Naples Parthenope, Naples, Italy, 7 Multidisciplinary Department of Medical, Surgical and Dental Sciences, Plastic Surgery Unit, University of Campania "L. Vanvitelli", Naples, Italy, 8 Cardiovascular Diseases Department, "Cardarelli Hospital", Naples, Italy

☯ These authors contributed equally to this work.
* concetta.schiano@unicampania.it

⊙ OPEN ACCESS

**Data Availability Statement:** All relevant data are within the paper and its Supporting Information files.

## Abstract

### Aims

Immune endothelial inflammation, underlying coronary heart disease (CHD) related phenotypes, could provide new insight into the pathobiology of the disease. We investigated DNA methylation level of the unique CpG island of *HLA-G* gene in CHD patients and evaluated the correlation with cardiac computed tomography angiography (CCTA) features.

### Methods

Thirty-two patients that underwent CCTA for suspected CHD were enrolled for this study. Obstructive CHD group included fourteen patients, in which there was a stenosis greater than or equal to 50% in one or more of the major coronary arteries detected; whereas subjects with Calcium (Ca) Score = 0, uninjured coronaries and with no obstructive CHD (no critical stenosis, NCS) were considered as control subjects (n = 18). For both groups, DNA methylation profile of the whole 5'UTR-CpG island of *HLA-G* was measured. The plasma soluble HLA-G (sHLA-G) levels were detected in all subjects by specific ELISA assay. Statistical analysis was performed using R software.

### Results

For the first time, our study reported that 1) a significant hypomethylation characterized three specific fragments (B, C and F) of the 5'UTR-CpG island (p = 0.05) of *HLA-G* gene in

**Funding:** This work was supported by PRIN2017F8ZB89 from "Italian Ministry of University and Research (MIUR)" (PI Prof Napoli) and Ricerca Corrente (RC) 2019 from "Italian Ministry of Health" (PI Prof. Napoli).; by grants GR-2016-02364785 from "Italian Ministry of Health" (PI Dr. Grimaldi).

**Competing interests:** The authors have declared that no competing interests exist.

CHD patients compared to control group; 2) the hypomethylation level of one specific fragment of 161bp (+616/+777) positively correlated with coronary Ca score, a relevant parameter of CCTA (p<0.05) between two groups evaluated and was predictive for disease severity.

## Conclusions

Reduced levels of circulating HLA-G molecules could derive from epigenetic marks. Epigenetics phenomena induce hypomethylation of specific regions into 5'UTR-CpG island of *HLA-G* gene in CHD patients with obstructive non critical stenosis vs coronary stenosis individuals.

## 1. Introduction

Coronary heart disease (CHD) shows a complex nature resulting from several interacting genetic/epigenetic risk factors, which are strongly affected by individual lifestyle [1,2]. Since genome-wide association studies (GWASs) present several limitations in explaining the genotype-phenotype relationship, major efforts have been made to design an epigenetic map able to bridge the gap between genome and environment providing novel useful non-invasive biomarkers for CHD [3–7]. In the last years, advanced network-oriented analysis unveiled specific molecular pathways underlying CHD-related endophenotypes, mainly endothelial inflammation, providing novel insight in disease pathobiology [8–12]. Some authors emphasized the key role of DNA methylation in regulating the human leukocyte antigen-G (HLA-G) gene expression involved in inflammatory-related pathways underlying CHD onset [13,14]. HLA-G belongs to nonclassical Ib antigen of the major histocompatibility complex (MHC) representing a crucial immune checkpoint for the maintenance of self-tolerance and modulation of innate immune response [15]. At molecular level, DNA methylation is a covalent binding of a methyl (-CH3) group to the $5^{th}$ position of cytosine residue in the CpG dinucleotides enriched in CpG islands, which range from 200bp to several Kb, and are generally located near active genes. Besides, changes in CpG island DNA methylation profiles may affect tissue-specific gene expression at transcriptional level leading to higher risk for cardiac injury already in the early phase of fetal development [2,4–5,16]. Our previous study has demonstrated that peripheral blood cells from CHD patients presented differential DNA methylation changes in targeted genomic segments correlating with gene expression and disease severity *vs* controls [4]. Therefore, we hypothesized that CpG island HLA-G DNA methylation profile in peripheral blood cells may provide putative useful non-invasive predictive biomarkers and drug targets, which may improve CHD precision medicine and personalized therapy [17–19]. Besides, our manuscript reports the first study aiming to investigate the methylation status of the single CpG island of HLA-G gene by using CHD patients that resulted positive to cardiac computed tomography angiography (CCTA). Along with a targeted DNA methylation map, we aim to find a putative association between HLA-G CpG island DNA methylation changes and specific CCTA parameters. Since CCTA is not invasive and offers a good diagnostic performance [20,21], the integrated analysis of plaque load changes and DNA methylation signatures affecting HLA-G gene regulation may represent a useful approach to improve risk stratification for CHD patients.

## 2. Methods

### 2.1 Power analysis

During the experimental design, a power analysis was performed using G*Power software. The minimum sample size was calculated with a priori power analysis comparing two groups (critical and non-critical stenosis) and using unpaired Mann-Whitney U test with a power of about 0.8, an alpha level of 0.05 and medium effect size (d = 0.7). Based on the assumptions, the minimum total sample size was 32.

### 2.2 Patient selection

The study was approved by the institutional ethics committee in accordance with the ethical guidelines of the 1975 Declaration of Helsinki and approved by ethical committee of IRCCS SDN (Protocol no. 7–13). A written informed consent was obtained from all subjects included in the study. At IRCCS SDN, during a period of one year, 90 consecutive patients were enrolled in the study. Patients with diagnosed history of malignancy disorders, active infections, chronic or immune-mediated diseases were excluded from the study to avoid confounding effects due to other variables. Moreover, subjects with cardiomyopathy, known CHD, previous percutaneous transluminal coronary angioplasty and coronary artery bypass grafting, systemic atherosclerosis, such as lower extremity peripheral arterial disease or supra-aortic arterial disease were not included in the study. After all these exclusion criteria, 32 subjects without a history of cardiovascular (CV) events and underwent to IRCCS SDN institution for suspected CHD, were considered for the study. Obstructive CHD was defined by the presence of a coronary stenosis ≥50% in one or more of the major coronary arteries detected by CCTA (n = 14). Patients with Calcium (Ca) Score = 0, uninjured coronaries and with no obstructive CHD (defined as the presence of a stenosis minor of 50% in one or more of the major coronary arteries) were considered as control patients (n = 18).

All clinical characteristics such as laboratory parameters, presence of cardiovascular risk factors, and medical history were accurately recorded. Dyslipidemia was defined as treatment with drugs or fasting serum total cholesterol ≥200 mg/dL, or LDL cholesterol ≥70 mg/dL, or high-density lipoprotein cholesterol <40 mg/dL, or triglyceride ≥150 mg/dL [22].

Hypertension was defined as treatment with drugs or systolic blood pressure (SBP)≥140 mmHg or diastolic blood pressure (DBP) ≥90 mmHg [23]. Anthropometrical measurements including body weight and height were recorded and body mass index (BMI) was calculated. Blood pressure and resting heart rate were measured after ≥ 5 min rest with a sphygmomanometer (Table 1).

### 2.3 Sample collection and molecular analysis

From all suspected patients, peripheral venous blood samples were collected in EDTA tubes after 6–8 hours fasting. All experimental procedures were performed at 4˚C within 1 hour of collection to separate plasma and cellular components. Peripheral blood mononuclear cells (PBMNCs) were isolated by Ficoll gradient using HISTOPAQUE-1077 (Sigma Diagnostics, USA) according to manufacturer's instructions and frozen at -80˚C at the IRCCS SDN Biobank.

### 2.4 CCTA and image analysis

All subjects underwent a CCTA by a third-generation dual source multidetector computed tomography scanner (Siemens Healthcare AG, Germany). After an ECG-triggered high pitch spiral acquisition (FLASH) without contrast medium was performed for calcium score

**Table 1. Baseline characteristics of patients with Non-Critical Stenosis (NCS) and Critical Stenosis (CS).**

| Variables | Control group (NCS<50%) (n = 18) | Critical Stenosis group (CS≥50%) (n = 14) | *p* value |
|---|---|---|---|
| Age (years)* | 55.11±10.62 | 61.78±11.87 | 0.11 |
| Gender (M/W) (M/W%) | 11/7 (61.1/38.9%) | 13/1 (92.9/7.1%) | **0.04** |
| BMI (kg/m²)* | 27.59 ±3.88 | 29.84 ±5.89 | 0.23 |
| Total Cholesterol (mg/dL)* | 190.33±33.07 | 187.20±46.29 | 0.90 |
| LDL Cholesterol (mg/dL)* | 121.00 ±20.16 | 126.75±41.66 | 0.81 |
| HDL Cholesterol (mg/dL)* | 50.4±6.188 | 44.0±5.70 | 0.13 |
| Pericardial fat (mL)* | 155.45±70.61 | 211.05±115.91 | 0.13 |
| CHD familiarity (%) | 10 (55.6%) | 7 (50.0%) | 0.75 |
| Smoking (%) | 5 (27.8%) | 4 (28.6%) | 0.96 |
| Diabetes (%) | 1 (5.6%) | 2 (14.3%) | 0.40 |
| Hypertension (%) | 15 (83.3%) | 8 (57.1%) | 0.10 |
| Dyslipidemias (%) | 6 (33.3%) | 5 (35.7%) | 0.89 |
| Physical Activity (%) | 4 (22.2%) | 3 (21.4%) | 0.96 |

*Data are represented as mean ± SD. Bold values were considered statistically significant with a $p < 0.05$.

*Abbreviations*: BMI: body mass index; CHD: coronary heart disease; HDL: high density lipoprotein; LDL: low density lipoprotein; M: man; W: woman.

evaluation (slice thickness of 3mm, increment of 3 mm, small FOV), an angiographic Cardiac CT scans with IV contrast material (Iodinated contrast agent—Iomeprol 400 mg I/ml—Iomeron 400) (Bracco, Italy), followed by saline flush, was performed. Then, scans were executed with retrospective ECG gating and with prospective ECG-tube current modulation (window 25%-75% of the R-R interval). Imaging parameters were restructured with a 3rd generation advanced modeled iterative reconstruction (ADMIRE, Siemens) with a strength level of 3 using different convolution kernels (Bv36, Bv40, Bv44 and Bv49) with the smallest FOV possible. CCTA derived features, such as Ca Score, stenosis degree, expressed in percentage of lumen reduction, plaque composition, specified as calcified, non-calcified, or mixed subtypes, and total number of plaque segments were investigated. A consensus interpretation was obtained according to the international SCCT guidelines [24]. A reduction in the luminal diameter of ≥50% in one or more of the major coronary arteries represented a critical coronary stenosis. In order to stratify patients with critical stenosis (CS) (≥50%) and non-critical stenosis (NCS) (<50%), stenosis degree was calculated.

## 2.5 Methylated DNA Immunoprecipitation (MeDIP)

For DNA extraction and immunoprecipitation from isolated PBMNCs was used MagMeDIP kit™ (Diagenode, Belgium). Genomic DNA was extracted, and 30 μg were sheared (10 cycles, 15 s "ON", 15 s "OFF" at 20% of amplitude) in fragments between 100–800 bp using the Q125 sonicator (Qsonica, USA). Shared DNA was analyzed on agarose gel. MeDIP was performed using α-5′methyl-cytosine antibody; samples were rotated overnight at 4°C in the presence of magnetic beads. The 10% of IP incubation mix of each DNA shared sample was stored as input for the comparison with immunoprecipitated DNA. Immunoprecipitated DNA was eluted in TE buffer. The amount of methylated DNA enrichment in MeDIP samples compared to the respective INPUT sample was detected by qRT-PCR CFX96 Touch Real-Time PCR Detection System (BioRad Laboratories, Ltd, USA) with iQ™ SYBR® Green Supermix (BioRad Laboratories, Ltd, USA). A set of specific primer pairs provided in the kit and targeting specific DNA sequences were used for checking MeDIP efficiency. The Methyl DNAIP controls revealed IP efficiency. qRT-PCR data were expressed as percentage of methylated DNA IP

**Table 2. DNA methylation specific primers for CpG island of HLAG gene.**

| Primer name identifier | Forward | Reverse | Product size (bp) | chr6 position |
|---|---|---|---|---|
| HLAG_A | GCGGTCCTGGTTCTAAAGTC | GAGAGTAGCAGGAAGAGGGT | 102 | 1096146+1096247 |
| HLAG_B | CTCTTCCTGCTACTCTCGGG | CTCATGGAGTGGGAGCCT | 194 | 1096231+1096424 |
| HLAG_C | ATGAGGTATTTCAGCGCCG | GTGTTCCGTGTCTCCTCTTC | 188 | 1096420+1096607 |
| HLAG_D | GAAGAGGAGACACGGAACAC | GGGGTTACTCACTGGCCT | 103 | 1096588+1096690 |
| HLAG_E | AGGCCAGTGAGTAACCCC | GCAGGGATTTTGGTAAAGGC | 172 | 1096673+1096844 |
| HLAG_F | GCCTTTACCAAAATCCCTGC | AGGCATACTGTTCATACCCG | 161 | 1096825+1096985 |
| HLAG_G | TCCGCGGGTATGAACAGTAT | CTCTCCTTTGTTCAGCCACA | 144 | 1096962+1097105 |
| HLAG_H | GTGGCTGAACAAAGGAGAGC | CTCAGGGTGGCCTCATAGTC | 151 | 1097087+1097837 |

compared to input (% of DNAIP/total input). Each sample was analyzed in triplicate and data expressed as mean ±standard error (SE).

Genome Browser tool (https://genome.ucsc.edu) were used to select genetic region and design DNA methylation specific primers for CpG island of *HLA-G* gene (Table 2).

## 2.6 ELISA assay

The determination of HLA-G soluble forms was performed by sandwich enzyme linked immunoassay (ELISA) kit (Exbio, Czech Republic) using microplates pre-coated with anti-sHLA-G monoclonal antibody. After 16–20 hours of incubation and repeated washing, the horseradish peroxidase-labeled human anti-β2-microglobulin monoclonal antibody (HRP) was added to all wells and incubated for 60 minutes with captured sHLA-G. The HRP conjugate reacted with the substrate solution (TMB). Absorbance was detected on the standard plate reader Infinite 200 PRO (Tecan Group Ltd., Switzerland). sHLA-G concentration was determined according to the standard curves and expressed as U/mL or ng/mL [25].

## 2.7 Statistical analysis

Statistical analysis was performed using R software (version 3.03, Austria). Continuous variables were expressed as mean ±standard deviation (SD) or standard error (SE). Data were tested for normality through the Shapiro-Wilk test. Unpaired Student's t-test or Mann-Whitney U test, as required, were used for comparison between two groups, CS and NCS. Categorical variables were expressed as percentage and were compared using the Chi-Square test or the Fisher's exact test. A p-value less than 0.05 was considered significant. Bonferroni's correction was used for multiple hypothesis correction if necessary.

We identified significant variation in HLA nucleotides methylation levels between the two groups (NCS vs CS) and then we investigated the prognostic power of the HLA-G statistically significant methylation levels for relative CpG islands in order to predict the severity of disease (CS). For this purpose, we considered a Generalized Linear Model (GLM) and we evaluated the performance using a 3-fold cross validation. Finally, the association between CCTA variables and significant HLA nucleotides methylation levels were investigated in CS group. Spearman's rank correlation was conducted for continuous CCTA variables and linear regression analysis was carried out for categorical CCTA variables. A Spearman's ρ or regression's R value greater than 0.8 and significant p-value (p-value<0.05) was set as threshold to identify strong agreement between CCTA parameters and HLA nucleotides methylation levels.

## 3. Results

### 3.1 Study population

Our manuscript reports the first study aiming to investigate the methylation status of the single 5'-CpG island of *HLA-G* gene by using Methylated DNA Immunoprecipitation (MeDIP) technique performed on peripheral blood mononuclear cells (PBMNCs) extracted from subjects that underwent CCTA for suspected CHD. A total number of 32 patients were enrolled in this study. Fourteen patients showed a critical stenosis (CS) ($\geq$50%), of which the 50% (n = 7) showed a critical stenosis of >75%, the remaining eighteen patients showed a stenosis degree <50% (NCS).

The mean age was 61.78±11.87 years in CS compared to 55.11±10.62 years in NCS group (p = 0.110). The percentage of male was significantly higher in patients with CS (92.9%) compared to NCS (61.1%, p = 0.041).

The mean body mass index (BMI), the pericardial fat and cardiovascular risk factors (CHD familiarity, smoke, diabetes, hypertension and dyslipidemia, total cholesterol, LDL- and HDL-cholesterol plasmatic concentrations) were comparable between two groups, but they were not significantly different between CS and NCS patients (Table 1).

### 3.2 CCTA

For all subjects imaging features were extracted by CCTA (Fig 1). Specifically, Ca score, plaque composition, specified as calcified, non-calcified, or mixed subtypes, total number of stenotic vessels and plaque segments were investigated between NCS and CS groups. Table 3 shows the results from the statistical analysis. All CCTA features changed significantly (p-value less than 0.05) between NCS and CS individuals. Mean (SD) values for Ca score and counts (%) for categorical CCTA features were zero for control patients (NCS).

### 3.3 Differential methylation of *HLA-G* 5'-CpG island

This pilot study allows us the analysis of unique 5'UTR-CpG island of gene coding for *HLA-G*. An epigenome analysis of DNA methylation from PBMCs was performed in n = 14 angiographically positive subjects (cases), who were age matched with n = 18 angiographically negative controls. Specifically, the cases reported a CS ($\geq$50%) at coronary level, whereas NCS showed a non-critical stenosis (<50%). By statistical analysis, significantly lower (Fig 2) (Table 4).

The distance of fragments was positioned from the ATG initiation codon of *HLA-G* gene. The amplified fragments B- and C-related regions were located into exon 1 and exon 2 respectively, whereas F region was in the second intron of the gene. In addition, we investigated whether significant CpG methylation levels of *HLA-G* gene (as reported in Table 4) could be considered as a prognostic value to predict the severity of disease associated to CS. The results showed that *HLA-G* methylation for F fragment reported the best performance in sensitivity and specificity values in order to predict CS condition (Table 5).

### 3.4 Correlation analysis between methylation of *HLA-G* 5'-CpG island and imaging parameters

The statistical analysis for the quantitative features showed that all CCTA variables, such as Ca score, plaque composition, number of stenotic vessels and number of plaque segments, changed significantly between NCS and CS patients (Fig 3) (Table 3). In addition, we performed a correlation analysis and linear regression in order to evaluate possible association

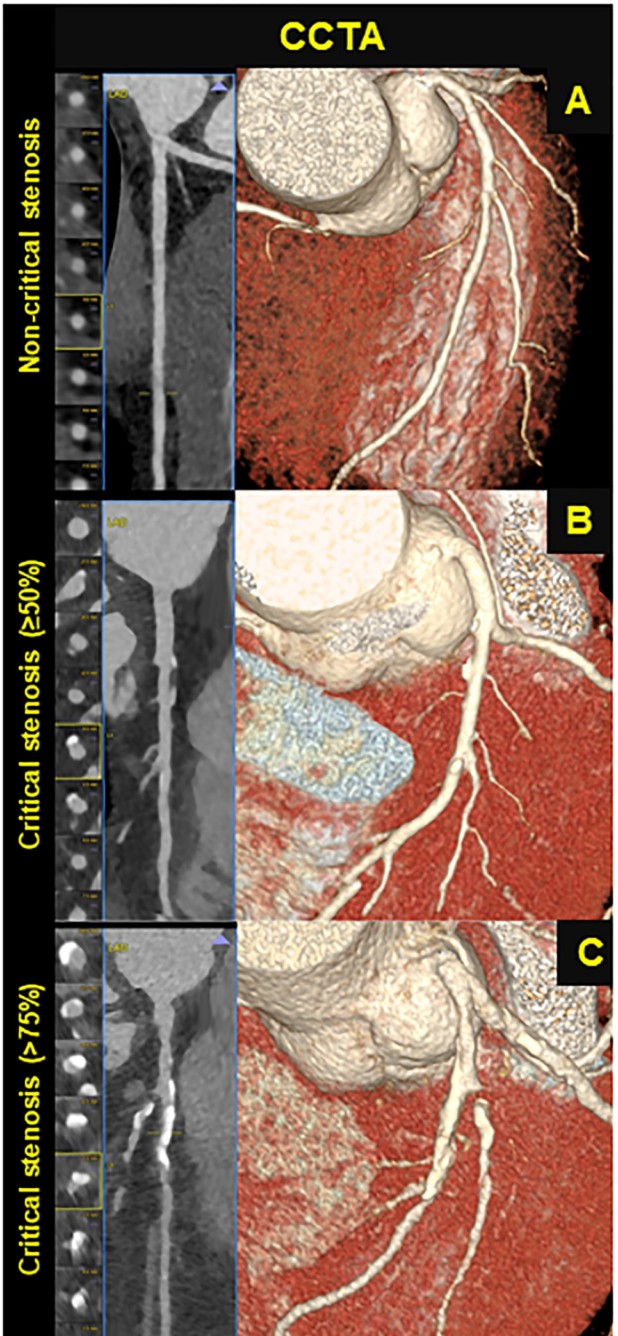

**Fig 1. CCTA screening.** CCTA analysis in subjects screened for suspected CHD. **Panel A** shows a subject with non-critical stenosis (<50); **Panel B** a patient with a critical stenosis ≥50%; **Panel C** a patient with a critical stenosis>75%.

between imaging and molecular features. The results of correlation analysis (S1 Table) showed a significant association of Ca Score with methylation level of the fragment F (+616/+777) in 5'-CpG region (Rho = 0.57, p = 0.03) as shown in Fig 4, whereas the regression analysis reported non-significant results (S1–S3 Figs).

**Table 3. Statistical analysis on quantitative imaging features.**

| Variables | NCS group (NCS<50%) | CS group (CS≥50%) | *p* value |
|---|---|---|---|
| **Ca score** | **0.14±0.27** | **550.03±600.17** | **0.004** |
| **Plaque composition** | | | **0.010** |
| Calcified | **0.00%** | **28.60%** | - |
| Non-calcified | **72.20%** | **28.60%** | - |
| Mixed | **27.80%** | **42.90%** | - |
| **N° Stenotic vessel** | **0.00±0.00** | **1.71±1.32** | **<0.001** |
| **N° plaque segments** | **1.27±0.46** | **5.42±4.07** | **0.002** |

Bold values were considered statistically significant with a p < 0.05.

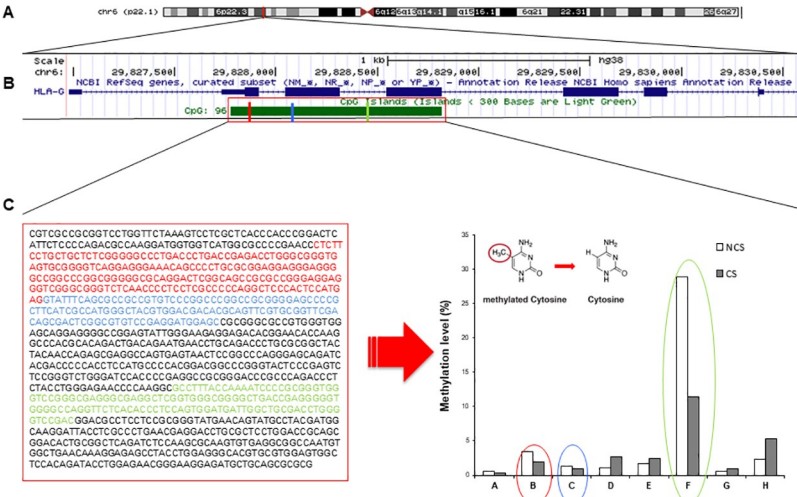

**Fig 2. *HLA-G* gene and relative 5'UTR-CpG island.** Graphical representation of *HLA-G* gene including its relative 5'UTR-CpG island. **A)** The physical map of the chr6q22.1, showing the genomic region encompassing *HLA-G* gene, is depicted. The 5'UTR-CpG island is also shown. **B)** Ref Seq annotated transcript is shown in blue. **C)** Methylation levels of all fragments analyzed are shown. Red, blue and green circles indicate a significant hypomethylation between NCS and CS group (p = 0.05).

**Table 4. Statistical analysis on molecular features.**

| Variables (%) | Control group (NCS<50%) (n = 18) | Critical Stenosis group (CS≥50%) (n = 14) | *p* value |
|---|---|---|---|
| HLAG_A_meth | 0.67±1.13 | 0.38±0.29 | 0.44 |
| **HLAG_B_meth** | **3.84±7.15** | **1.88±2.94** | **0.05** |
| **HLAG_C_meth** | **1.29±1.26** | **0.89±0.94** | **0.05** |
| HLAG_D_meth | 1.19±0.75 | 2.54±5.28 | 0.31 |
| HLAG_E_meth | 1.80±2.32 | 2.28±5.30 | 0.09 |
| **HLAG_F_meth** | **13.34±3.92** | **11.59±7.07** | **0.05** |
| HLAG_G_meth | 0.65±1.13 | 0.86±2.19 | 0.19 |
| HLAG_H_meth | 2.65±3.53 | 4.60±6.33 | 0.720 |

Bold values were considered statistically significant with a p < 0.05.

**Table 5. Model performance for relative 5'UTR-CpG island methylation levels of *HLA-G* gene with statistic significant changes between NCS and CS condition.**

| | HLAG_B | HLAG_C | HLAG_F |
|---|---|---|---|
| **Sensitivity** | 0% | 23% | *57%* |
| **Specificity** | 100% | 100% | *89%* |
| **Pos Pred Value** | NaN | 100% | *80%* |
| **Neg Pred Value** | 56% | 63% | *73%* |
| **Precision** | NaN | 100% | *80%* |
| **Recall** | 0% | 23% | *57%* |
| **Balanced Accuracy** | 50% | 62% | *73%* |

*NaN not available number

## 3.5 ELISA

All subjects were evaluated for sHLA-G plasma levels. CS group showed higher sHLA-G level compared to NCS, although not significant (p = 0.30) (Fig 5A). Plasma levels of sHLA-G were included between 0.0–6.5 U/mL (0.0–23.3 ng/mL) for the NCS group, whereas values included between 0.0–13.9 U/mL (0.00–50.0 ng/mL) were observed for CS group. We compared sHLA-G protein distribution with the significant distribution of the relative 5'UTR-CpG islands (HLAG_B, HLAG_C and HLAG_F) reported in Table 5, grouped by NCS and CS patients. The results showed an increase in circulating plasma protein for CS group compared to NCS group (Fig 5A), which corresponded to a decrease in 5'UTR-CpG methylation levels (Fig 5B). We performed a correlation analysis between methylation levels and circulating plasma protein for both groups, using Spearman's correlation coefficient. Unfortunately, the results did not report statistical significance.

## 4. Discussion

Our study reported: 1) a status of significant hypomethylation characterizing three specific fragments of the unique 5'-UTR CpG island (B, C and F) in *HLA-G* gene of control vs CHD patients with CS≥50%; 2) a strong correlation of hypomethylation levels of the F fragment (+616/+777) and coronary Ca score between controls and CHD with CS≥50% groups (Rho = 0.57, p = 0.03).

The clinical role of *HLA-G* has been widely explored in various pathophysiological conditions, such as organ transplantation, viral infection, autoimmune and inflammatory diseases, and cancer [13,14,26–29].

Moreover, in the literature, it is already known that *HLA-G* expression is modified by several genetic polymorphisms (SNPs)and some reports showed the association between SNPs and CHD risk although there is no evidence at transcriptomic level [30–32]. SNPs in distinct regions can play a relevant role both at transcriptional and protein levels, influencing binding of specific microRNAs and relative gene expression.

It was reported that *HLA-G* is upregulated in response to rejection of organ transplantation [33–37], and heart failure (HF) [38], suggesting a putative role to modulate the inflammatory condition [39,40].

Remarkable, in the literature, there are no studies about correlation between *HLA-G* methylation of 5'UTR-CpG island and imaging features. Thus, it was interesting to investigate in detail its regulation impact in CHD patients.

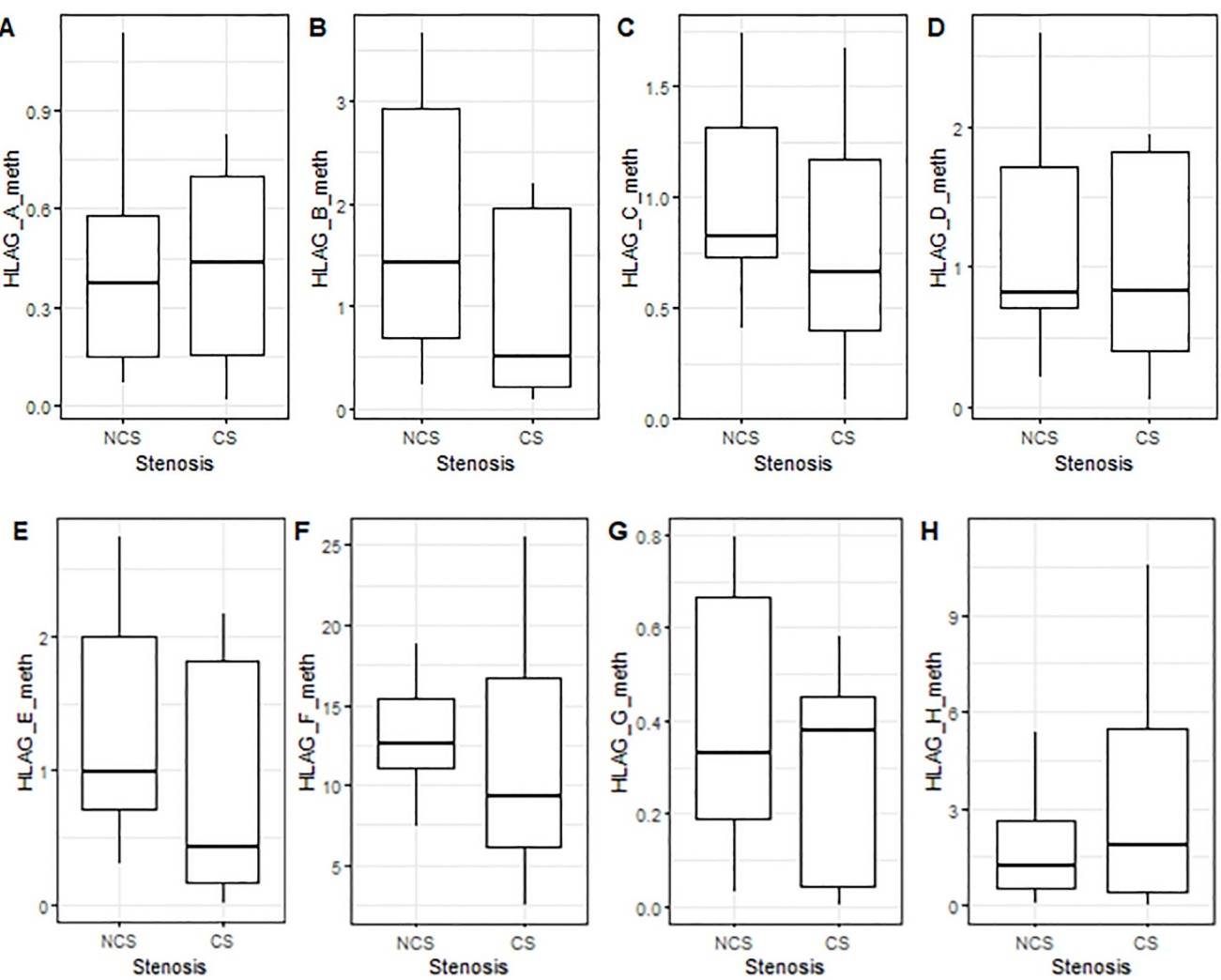

**Fig 3. Methylation levels of HLA-G CpG island.** Methylation levels of 5'-CpG HLA-G, evaluated by specific oligonucleotides, and gene expression in NCS and CS patients. **(A-I)** HLA-G oligonucleotides methylation levels; **(J-L)** gene expression. NCS: Non-critical Stenosis; CS: Critical Stenosis.

Recently, a pilot study analyzed DNA methylome by including 6 angiographically positive CHD patients compared to 6 controls. Results suggested that hypomethylation level of a specific CpG (cg06316104) of *HLA-G* promoter may be a useful predictive biomarker of disease [41]. Here, the authors performed an epigenome-wide analysis of methylome by using the microarray chip having a limited coverage of CpG sites. In our study, for the first time, we evaluated the differentially methylated regions (DMRs) considering the complete 5'UTR-CpG island of *HLA-G*. The mRNA synthesis of a specific gene is mainly regulated by the 5'UTR promoter region of the relative gene, as well as by the degree of degradation, localization and translation of the specific mRNA [42]. In our study, whole blood sample was chosen as the starting material method, since it is the least invasive.

It was reported that the epigenetic mechanisms could regulate gene-environment relationship underling individual responsiveness to CHD onset [43,44],For the first time, in the 1997, *Onno et al.*, reported a status of hypomethylation only in random six fragments of 5'-CpG island of *HLA-G G* in peripheral blood [45]. By analyzing the entire 5'UTR-CpG island, we observed that some DMRs were hypomethylated in accordance with prior discovery [45].

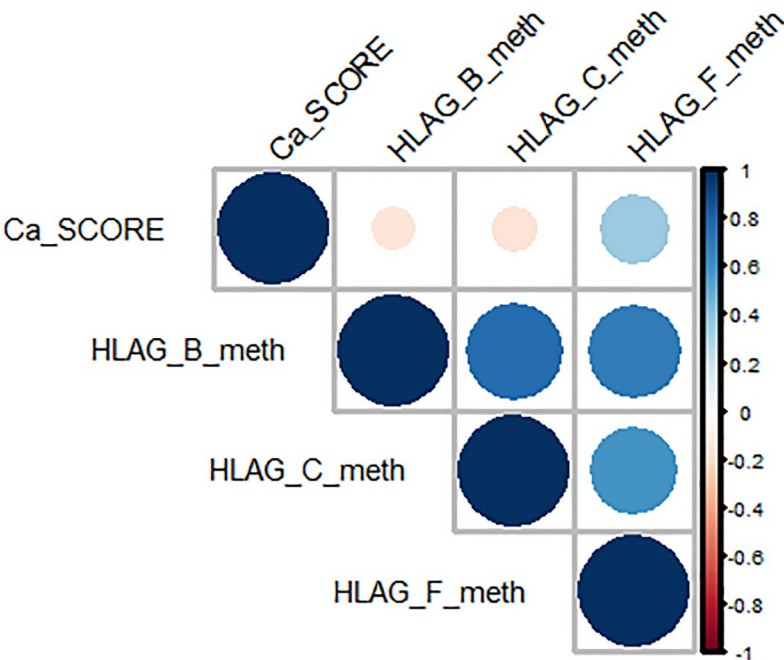

**Fig 4. 5'Cp*GHLA-G* methylation and Ca score correlation.** Correlation plot for Ca score continuous variables and HLA-G nucleotides methylation levels (B, C, F).

Particularly, 4 DMRs were hyper-, whereas 4 DMRs were hypomethylated. Moreover, we observed a significant reduction of methylation level of B(+22/+216), C (+211/+399) and F (+616/+777) fragments specifically in CHD patients with CS≥50% respect to NCS<50% ($p$<0.05). Notable, the regions relative to B- and C-oligonucleotides amplification were located into exon 1 and exon 2 respectively, whereas F region was positioned in the intron 2 of *HLA-G* gene.

In according to *Golareh et al.* results [46], in order to evaluate the potential role of DNA methylation to predict CHD risk, we performed a predictive statistical model on

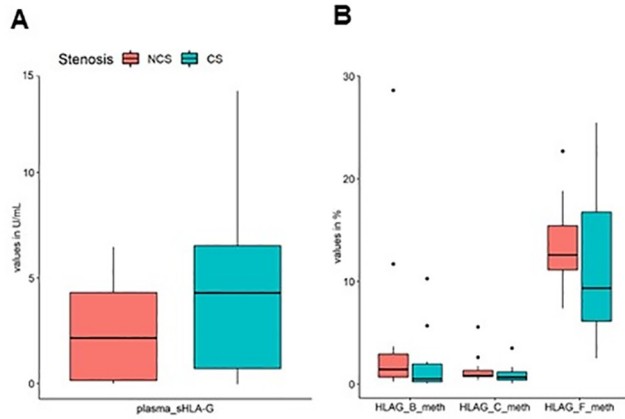

**Fig 5. Evaluation of plasmatic sHLA-G and relative CpGs methylation levels.** sHLA-G levels were analyzed by ELISA assay in the plasma samples of NCS and CS patients. **A)** Levels of plasmatic sHLA-G expressed as ng/mL; **B)** HLA-G methylation levels for relative 5'UTR-CpG-B,—C and -F were reported.

hypomethylated DMRs and we evaluated good performances considering 5'UTR-CpG island for F fragment characterizing disease severity in term of CS.

The statistical analysis for the quantitative features showed that all CCTA variables, such as Ca score, plaque composition, number of stenotic vessels and number of plaque segments, changed significantly between NCS and CS patients. In addition, in order to evaluate possible association between imaging and molecular features, a correlation analysis and linear regression were performed. The results of correlation analysis showed a significant association of Ca Score with methylation level of the F fragment(+616/+777) in 5'UTR-CpG region of *HLA-G* (p = 0.03) as shown in Fig 3, whereas the regression analysis reported no significant results (S1–S3 Figs). Finally, in order to understand whether epigenetic modifications could regulate circulating levels of HLA-G molecules, we detected plasma concentrations of sHLA-G in all subjects. To date, there are mixed data regarding its circulating levels during inflammatory diseases, such as rheumatoid arthritis (RA), Chron's disease (CD) and systemic lupus erythematosus (SLE) [47–49]. Most recently, it was demonstrated that sHLA-G levels were significantly lower in RA patients respect to healthy subjects [47];*Rizzo et al.* showed a higher secretion of sHLA-G in CD patients [48]; no significant difference was detected between plasma levels of SLE patients and control group, although a reduced plasma concentration of sHLA-G was observed when the patients were stratified according to clinical manifestations [49]. Regarding CVDs, higher levels of circulating sHLA-G were observed in heart transplantation [50], whereas there are no data about sHLA-G concentrations in CHD patients. In our study, for all subjects, plasma levels of sHLA-G were evaluated, but a not statistically significant difference of sHLA-G level was detected into CS group compared to NCS individuals (p = 0.09). In according to recent study reporting that changes in DNA methylation states are also associated with CVD pathophysiology [51], we evaluated that the decreasing in *HLA-G* methylation levels of relative 5'UTR-CpG island corresponded to the increasing of circulating sHLA-G protein concentration. The principal limitation of this study was the small sample size. Therefore, after this pilot study, which helped us to establish an analysis framework, the next step will include a clinically larger and more recent dataset to verify and validate these preliminary results in order to investigate possible correlation between imaging and molecular features involving in CHD.

## Supporting information

**S1 Table. Correlation analysis between CCTA variables and methylation levels on quantitative imaging features.**
(DOCX)

**S1 Fig. Plaque composition.** Regression Line for Plaque composition. **(A-C)** HLA-G nucleotides methylation levels (B, C, F) against the plaque composition variables.
(TIFF)

**S2 Fig. Number of Plaque segments.** Regression Line for Number of Plaque segments. **(A-C)** HLA-G nucleotides methylation levels (B, C, F) against the number of plaque segments.
(TIFF)

**S3 Fig. Number of stenotic vessels.** Regression Line for Number of stenotic vessels. **(A-C)** HLA-G nucleotides methylation levels (B, C, F) against the number of stenotic vessels.
(TIFF)

## Acknowledgments

We thank Dr. Linda Sommese (University of Campania "L. Vanvitelli") for her suggestions during study design and early stages of experimental procedures.

## Author Contributions

**Conceptualization:** Concetta Schiano, Giuditta Benincasa.

**Data curation:** Concetta Schiano.

**Formal analysis:** Concetta Schiano.

**Funding acquisition:** Marco Salvatore, Claudio Napoli.

**Investigation:** Carmela Fiorito.

**Methodology:** Giuditta Benincasa, Teresa Infante, Vincenzo Grimaldi.

**Project administration:** Concetta Schiano, Claudio Napoli.

**Software:** Monica Franzese, Rossana Castaldo.

**Supervision:** Gelsomina Mansueto, Gerardo Fatone, Andrea Soricelli, Giovanni Francesco Nicoletti, Antonio Ruocco, Ciro Mauro, Claudio Napoli.

**Visualization:** Andrea Soricelli, Giovanni Francesco Nicoletti, Antonio Ruocco, Ciro Mauro, Marco Salvatore, Claudio Napoli.

**Writing – original draft:** Concetta Schiano.

**Writing – review & editing:** Giovanni Della Valle.

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
