## [Decision Letter · Decision Letter 0]

28 Jun 2020

PONE-D-20-07217

Integrated analysis of DNA methylation profile of HLA-G gene and imaging in coronary heart disease: pilot study

PLOS ONE

Dear Dr. Schiano,

Thank you for submitting your manuscript to PLOS ONE. After careful consideration, we feel that it has merit but does not fully meet PLOS ONE’s publication criteria as it currently stands. Therefore, we invite you to submit a revised version of the manuscript that addresses the points raised during the review process.

We look forward to receiving your revised manuscript.

Kind regards,

Osman El-Maarri, Ph.D

Academic Editor

PLOS ONE

Journal Requirements:

Reviewers' comments:

Reviewer's Responses to Questions

**Comments to the Author**

1. Is the manuscript technically sound, and do the data support the conclusions?

Reviewer #1: Yes

Reviewer #2: Partly

2. Has the statistical analysis been performed appropriately and rigorously? 

Reviewer #1: Yes

Reviewer #2: Yes

3. Have the authors made all data underlying the findings in their manuscript fully available?

Reviewer #1: Yes

Reviewer #2: No

4. Is the manuscript presented in an intelligible fashion and written in standard English?

Reviewer #1: Yes

Reviewer #2: Yes

5. Review Comments to the Author

Reviewer #1: Interesting paper and results, with the limitations of a pilot study. It will be interesting to see if this test can be used to predict severity of disease and evaluate use evaluating prognostic value.

Reviewer #2: This study investigated DNA methylation level of the unique CpG island of HLA-G gene in CHD patients and evaluated the correlation with cardiac computed tomography angiography (CCTA) features. It is intresting and of clinical significance. However, there are some aspects need to be improved.

1.In the section of in Introduction, the author should express why choose the DNA methylation as the study Aims and write the passage more logistic.

2.Table 1 in the “2.2 Patient selection” can be deleted， for it is same as table1 in the result.

3.For gender, non-critical stenosis (NCS) and critical stenosis (CS) was diffirently, the auther need to take subgroup analysis in the section of comparing DNA methylation leveland evaluated the correlation with cardiac computed tomography angiography (CCTA) features

4.Table 3 , the diffirence methylation level between non-critical stenosis (NCS) and critical stenosis (CS) was close to each other， so the result should be taken cautiously.

5.Table 3, the author need to add Control group

6. The author should add the analysis of mRNA expression, the investigate the relationship between mRNA expression and methylation level

6. PLOS authors have the option to publish the peer review history of their article (what does this mean?). If published, this will include your full peer review and any attached files.

Reviewer #1: No

Reviewer #2: No

---

## [Author Response · Author response to Decision Letter 0]

2 Jul 2020

PONE-D-20-07217

Integrated analysis of DNA methylation profile of HLA-G gene and imaging in coronary heart disease: pilot study

 Comments Answers

Reviewer #1:

 Interesting paper and results, with the limitations of a pilot study. 

It will be interesting to see if this test can be used to predict severity of disease and evaluate use evaluating prognostic value.

 We thank the reviewer for this comment. 

We agreed with the reviewer. Therefore, we investigated the prognostic value of the statistically significant methylation levels to predict the severity of disease (critical stenosis) via Generalized linear model (GLM). 

The performance was evaluated using a 3-fold cross validation. The best performance was achieved using 5’UTR-CpG of HLAG_F methylation level. Data were reported in a new Table (Table 5).

Reviewer #2:

 This study investigated DNA methylation level of the unique CpG island of HLA-G gene in CHD patients and evaluated the correlation with cardiac computed tomography angiography (CCTA) features. It is interesting and of clinical significance.

 We thank the reviewer for these comments. 

 1. In the section of in Introduction, the author should express why choose the DNA methylation as the study Aims and write the passage more logistic.

 In according with this suggestion, in order to clarify the choice of DNA methylation as first analysis approach, we re-paraphrased Introduction section. 

 2. Table 1 in the “2.2 Patient selection” can be deleted， for it is same as table1 in the result. As suggested, we preferred to insert the Table titled “Baseline Characteristics of patients with non-critical stenosis (NCS) and critical stenosis (CS)” only in the Methods and rename it in the Results section. 

 3. For gender, non-critical stenosis (NCS) and critical stenosis (CS) was differently, the authors need to take subgroup analysis in the section of comparing DNA methylation level and evaluated the correlation with cardiac computed tomography angiography (CCTA) features

 We agree with the reviewer. However, due to the limitation of the pilot study, a subgroup analysis was not possible at this moment. As reported in Table 1, in the critical stenosis group only one female is included. Therefore, it was not possible to evaluate the correlation in this group. 

 1. Table 3, the difference methylation level between non-critical stenosis (NCS) and critical stenosis (CS) was close to each other， so the result should be taken cautiously. 

 We agree with the review and this aspect was better highlighted in the limitation of the study. 

 2. Table 3, the authors need to add Control group In according with the reviewer, we decided to include Control group (NCS) also if several values for imaging parameters were about zero. We added this column in the Table 3 for the final version of the text, as suggested.

 3. The author should add the analysis of mRNA expression, the investigate the relationship between mRNA expression and methylation level

 We thank the reviewer for this suggestion. Unfortunately, we have disposable only plasma, but no RNA samples for these patients. In order to validate our methylation gene results, we decided to analyze HLA-G plasma concentration levels, as already previously reported in the manuscript. In addition, here, we calculated the relationship between methylation grade and plasma concentrations. Finally, statistically significant methylation levels were investigated in a graphical inspection via Boxplot. 

The new Figure (Figure 5) was reported in the final version of the manuscript. 

We thank the Editor and Reviewers for their valuable comments. We have accepted all suggested comments allowing us to improve the text of our Original Article. Also, we have checked all text, also re-modulating the section of Introduction and introduced a new Figure (called Figure 5) and a new Table (called Table 5). The revised form was prepared keeping in mind the guidelines of the “PLOS ONE”. 

We have described the changes, which we have highlighted in yellow in the “Marked version”. 

Reviewer #1: 

Interesting paper and results, with the limitations of a pilot study. 

We thank the reviewer for this comment.

It will be interesting to see if this test can be used to predict severity of disease and evaluate use evaluating prognostic value.

We agreed with the reviewer. Therefore, we investigated the prognostic value of the statistically significant methylation levels to predict the severity of disease (critical stenosis) via Generalized linear model (GLM). The performance was evaluated using a 3-fold cross validation. The best performance was achieved using 5’UTR-CpG of HLAG_F methylation level. Data were reported in a new Table (Table 5).

Reviewer #2:

This study investigated DNA methylation level of the unique CpG island of HLA-G gene in CHD patients and evaluated the correlation with cardiac computed tomography angiography (CCTA) features. It is interesting and of clinical significance.

We thank the reviewer for these comments. 

1. In the section of in Introduction, the author should express why choose the DNA methylation as the study Aims and write the passage more logistic.

In according with this suggestion, in order to clarify the choice of DNA methylation as first analysis approach, we re-paraphrased Introduction section. 

2. Table 1 in the “2.2 Patient selection” can be deleted， for it is same as table1 in the result. 

As suggested, we preferred to insert the Table titled “Baseline Characteristics of patients with non-critical stenosis (NCS) and critical stenosis (CS)” only in the Methods and rename it in the Results section. 

3. For gender, non-critical stenosis (NCS) and critical stenosis (CS) was differently, the authors need to take subgroup analysis in the section of comparing DNA methylation level and evaluated the correlation with cardiac computed tomography angiography (CCTA) features

We agree with the reviewer. However, due to the limitation of the pilot study, a subgroup analysis was not possible at this moment. As reported in Table 1, in the critical stenosis group only one female is included. Therefore, it was not possible to evaluate the correlation in this group. 

4. Table 3, the difference methylation level between non-critical stenosis (NCS) and critical stenosis (CS) was close to each other， so the result should be taken cautiously. 

We agree with the review and this aspect was better highlighted in the limitation of the study. 

5. Table 3, the authors need to add Control group.

In according with the reviewer, we decided to include Control group (NCS) also if several values for imaging parameters were about zero. We added this column in the Table 3 for the final version of the text, as suggested.

6. The author should add the analysis of mRNA expression, the investigate the relationship between mRNA expression and methylation level

We thank the reviewer for this suggestion. Unfortunately, we have disposable only plasma, but no RNA samples for these patients. In order to validate our methylation gene results, we decided to analyze HLA-G plasma concentration levels, as already previously reported in the manuscript. In addition, here, we calculated the relationship between methylation grade and plasma concentrations. Finally, statistically significant methylation levels were investigated in a graphical inspection via Boxplot. The new Figure (Figure 5) was reported in the final version of the manuscript.

---

## [Decision Letter · Decision Letter 1]

17 Jul 2020

Integrated analysis of DNA methylation profile of HLA-G gene and imaging in coronary heart disease: pilot study

PONE-D-20-07217R1

Dear Dr. Schiano,

We’re pleased to inform you that your manuscript has been judged scientifically suitable for publication and will be formally accepted for publication once it meets all outstanding technical requirements.

Kind regards,

Osman El-Maarri, Ph.D

Academic Editor

PLOS ONE

Additional Editor Comments (optional):

Reviewers' comments:

Reviewer's Responses to Questions

**Comments to the Author**

1. If the authors have adequately addressed your comments raised in a previous round of review and you feel that this manuscript is now acceptable for publication, you may indicate that here to bypass the “Comments to the Author” section, enter your conflict of interest statement in the “Confidential to Editor” section, and submit your "Accept" recommendation.

Reviewer #1: All comments have been addressed

Reviewer #2: All comments have been addressed

2. Is the manuscript technically sound, and do the data support the conclusions?

Reviewer #1: Yes

Reviewer #2: Yes

3. Has the statistical analysis been performed appropriately and rigorously? 

Reviewer #1: Yes

Reviewer #2: Yes

4. Have the authors made all data underlying the findings in their manuscript fully available?

Reviewer #1: Yes

Reviewer #2: Yes

5. Is the manuscript presented in an intelligible fashion and written in standard English?

Reviewer #1: Yes

Reviewer #2: Yes

6. Review Comments to the Author

Reviewer #1: Thank you for addressing the review questions - I have no more comments xxxxxxxxxxxxxxxxxxxxxxxxxxx

Reviewer #2: the authors have adequately addressed the comments raised in a previous round of review and this manuscript is now acceptable for publication

7. PLOS authors have the option to publish the peer review history of their article (what does this mean?). If published, this will include your full peer review and any attached files.

Reviewer #1: No

Reviewer #2: No